# Increase in global emissions of HFC-23 despite near-total expected reductions

K.M. Stanley [1,2]*, D. Say [1], J. Mühle [3], C.M. Harth[3], P.B. Krummel[4], D. Young [1], S.J. O'Doherty [1], P.K. Salameh[3], P.G. Simmonds[1], R.F. Weiss [3], R.G. Prinn[5], P.J. Fraser[4] & M. Rigby [1]*

Under the Kigali Amendment to the Montreal Protocol, new controls are being implemented to reduce emissions of HFC-23 ($CHF_3$), a by-product during the manufacture of HCFC-22 ($CHClF_2$). Starting in 2015, China and India, who dominate global HCFC-22 production (75% in 2017), set out ambitious programs to reduce HFC-23 emissions. Here, we estimate that these measures should have seen global emissions drop by 87% between 2014 and 2017. Instead, atmospheric observations show that emissions have increased and in 2018 were higher than at any point in history (15.9 ± 0.9 Gg $yr^{-1}$). Given the magnitude of the discrepancy between expected and observation-inferred emissions, it is likely that the reported reductions have not fully materialized or there may be substantial unreported production of HCFC-22, resulting in unaccounted-for HFC-23 by-product emissions. The difference between reported and observation-inferred estimates suggests that an additional ~309 Tg $CO_2$-equivalent emissions were added to the atmosphere between 2015 and 2017.

---

[1] School of Chemistry, University of Bristol, Bristol, UK. [2] Institute for Atmospheric and Environmental Science, Goethe University Frankfurt, Frankfurt, Germany. [3] Scripps Institution of Oceanography, University of California, San Diego, La Jolla, CA, USA. [4] Climate Science Centre, CSIRO Oceans and Atmosphere, Aspendale, VIC, Australia. [5] Center for Global Change Science, Massachusetts Institute of Technology, Cambridge, MA, USA. *email: stanley@iau.uni-frankfurt.de; Matt.Rigby@bristol.ac.uk

D ue to their influence on climate, the major hydrofluorocarbons (HFCs) were regulated under the Kyoto Protocol of the United Nations Framework Convention on Climate Change (UNFCCC) and have recently been included in the Kigali Amendment to the Montreal Protocol to ensure that their radiative forcings does not offset climate gains provided by the phaseout of chlorofluorocarbons (CFCs) and hydrochlorofluorocarbons (HCFCs)[1–3]. With a long atmospheric lifetime (~228 years[1,4]), and high radiative efficiency, HFC-23 has the highest global warming potential (GWP) among HFCs (100-year GWP, 12,690[1]). It is primarily produced as an unwanted by-product during the manufacture of the refrigerant HCFC-22, via the over-fluorination of chloroform ($CHCl_3$)[5]. Smaller emissions of HFC-23 are associated with its use as a feedstock for halon-1301 ($CBrF_3$) production, plasma etching and chamber cleaning in the semiconductor industry, very low-temperature refrigeration and specialty fire suppression[1,5–7].

Previous studies, based on in situ atmospheric observations and firn air measurements, have shown an increase in the global annual mean mole fraction of HFC-23 from near zero in early 1960 to $28.9 \pm 0.6$ pmol mol$^{-1}$ by the end of 2016[1,5,6]. These data, when combined with a model of atmospheric transport and chemistry can be used to infer global emissions. Such top-down methods have previously shown that global HFC-23 emissions grew from $4.2 \pm 0.7$ Gg yr$^{-1}$ in 1980 to $13.3 \pm 0.8$ Gg yr$^{-1}$ in 2006, before declining rapidly to $9.6 \pm 0.6$ Gg yr$^{-1}$ in 2009 in response to emission reductions from developed countries and as a result of the UNFCCC Clean Development Mechanism (CDM)[6,8]. As the CDM period ended, HFC-23 emissions grew to $14.5 \pm 0.6$ Gg yr$^{-1}$ in 2014, before declining again, to $12.7 \pm 0.6$ Gg yr$^{-1}$, in 2016[6].

Here, we present an update of global HFC-23 emissions derived from atmospheric observations, based on new data from 2015 to the end of 2018. In addition, we compile a new inventory-based HFC-23 emissions estimate through to the end of 2017 that includes reported emission reductions by China and India. We find that in 2018, observation-based HFC-23 emissions are higher than at any point in history ($15.9 \pm 0.9$ Gg yr$^{-1}$), whilst inventory-based emissions are at the lowest in the past 17 years ($2.4 \pm 0.9$ Gg yr$^{-1}$ in 2017) when reported emission reductions were included. Due to the magnitude of the discrepancy between reported emissions reductions and emissions inferred from the atmospheric data, it is highly likely that developing countries have been unsuccessful in meeting their reported emissions reductions. Alternatively, or additionally, there may be substantial unreported production of HCFC-22 at unknown locations resulting in unaccounted-for HFC-23 by-product being vented to the atmosphere.

## Results

**Bottom-up global HFC-23 emissions.** Inventory-based (bottom-up) estimates of global HFC-23 emissions can be derived from emissions reported to the UNFCCC and from reports of HCFC-22 production submitted to the United Nations Environment Programme (UNEP), combined with emissions factors (i.e. emissions to the atmosphere of HFC-23 per tonne of HCFC-22 produced).

We compiled a developing country no abatement bottom-up HFC-23 emissions estimate (Supplementary Table 1) based on HCFC-22 production data obtained from the UNEP HCFC database (https://ozone.unep.org/countries) (Supplementary Table 2) and available time-varying emissions factors[6,9–11] for developing nations (defined here as Article 5 countries under the Montreal Protocol and non-Annex I under the UNFCCC, including Israel). For developed countries (non-Article 5 countries under the Montreal Protocol and Annex I under the UNFCCC, including Turkey but excluding Israel), HFC-23 emission estimates were taken from the 2019 National Inventory

Reports (NIR) submitted to the UNFCCC[12] (Supplementary Table 1). Next, we compiled a developing country with abatement bottom-up estimate (Supplementary Table 1), which also includes reported HFC-23 emission reductions from the UNFCCC CDM (compiled data in Supplementary Table 3) between 2006 and 2014, whereby developing nations could provide Certified Emission Reduction (CER) credits for the destruction of HFC-23 by-product, which were then traded with developed countries to meet their emission reduction targets[1,6]. The level of abatement of HFC-23 reported under the CDM dropped and reached zero by the end of 2014. After 2015, expected abatement from developing countries is dominated by reported emissions reduction by China and India[9,10,13]. Under the Chinese HCFC production phase-out management plan (HPPMP), China reported a reduction of 45, 93 and 98% of total HFC-23 emissions in 2015, 2016 and 2017, respectively[10]. In India, an executive order (Control of emission/venting of Hydrofluorocarbon (HFC)-23, produced as by product while manufacturing of Hydrochlorofluorocarbon (HCFC)-22, in the atmosphere)[14] issued by the Indian government on 13 October 2016 required all producers of HCFC-22 to destroy by-product HFC-23 via incineration using efficient and proven technologies[9]. For simplicity, we collectively referred to these measures as post-2015 abatement.

Total HCFC-22 production has increased substantially between 1990 (65 Gg yr$^{-1}$) and 2017 (947 Gg yr$^{-1}$; Fig. 1b). Since 2005, HCFC-22 production has been dominated by developing countries. From 2013, a freeze in HCFC production for dispersive use has been in place under the Montreal Protocol[15]. However, a small increase from developing countries has been reported, from 675 Gg yr$^{-1}$ in 2013 to 725 Gg yr$^{-1}$ in 2017. For developed countries, HCFC production and consumption are due to be completely phased-out by 2030, with a 99.5% reduction expected by 2020 from the baseline year (1989)[16]. Notwithstanding this schedule, a small increase in total production has been reported from developed countries, from 208 Gg yr$^{-1}$ in 2016 to 222 Gg yr$^{-1}$ in 2017. Similarly, HFC-23 emissions reported to the UNFCCC (2019 NIR; aggregated values in Supplementary Table 1) show increases from 1.0 Gg yr$^{-1}$ in 2016 to 1.8 Gg yr$^{-1}$ in 2017. This increase in emissions is driven by fluorochemical production, predominantly from Russia (0.6 Gg yr$^{-1}$ in 2016 to 1.2 Gg yr$^{-1}$ in 2017) and the USA (0.2 Gg yr$^{-1}$ in 2016 to 0.4 Gg yr$^{-1}$ in 2017), possibly signalling a recent increase in production for non-dispersive HCFC-22 uses.

China and India, the two largest current producers of HCFC-22, with 2017 reported production of 645 Gg (68% of the global total) and 65 Gg (7%), respectively, have reported actions to dramatically reduce their emissions[9,10,14]. In the period between the cessation of Indian CDM projects (end of 2013) and 13 October 2016, Indian producers of HCFC-22 were allowed to vent HFC-23 by-product to the atmosphere. Indeed, emissions inferred from aircraft observations during the summer of 2016 suggest that a substantial fraction of India's HCFC-22 production-related emissions were likely unabated at that time[13]. Collectively, the post-2015 abatement reported by both countries is shown in the cross-hatched area in Fig. 1a. We estimate that the HFC-23 emissions reductions reported under China's HPPMP are 6.1, 13.0 and 15.2 Gg yr$^{-1}$ in 2015, 2016 and 2017, respectively, whilst India's reductions correspond to 0, 0.4 and 1.9 Gg yr$^{-1}$ for these years (Fig. 1a). Therefore, combined emission reductions should have totalled 6.1, 13.4 and 17.1 Gg yr$^{-1}$ in 2015, 2016 and 2017, respectively, leading to a global total emission rate of 2.4 Gg yr$^{-1}$ by 2017.

**Evaluating global emissions using atmospheric observations.** We provide an update on global HFC-23 emissions through to the end of 2018, based on in situ HFC-23 measurements from the

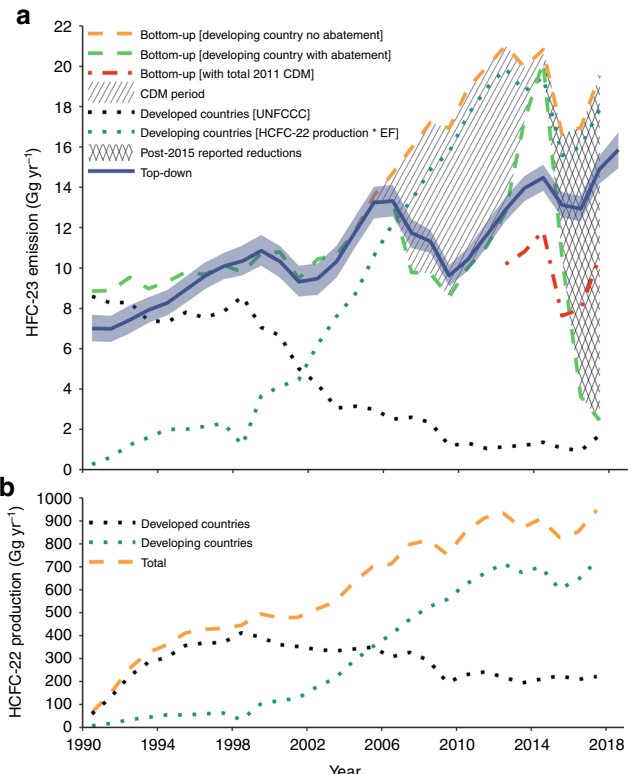

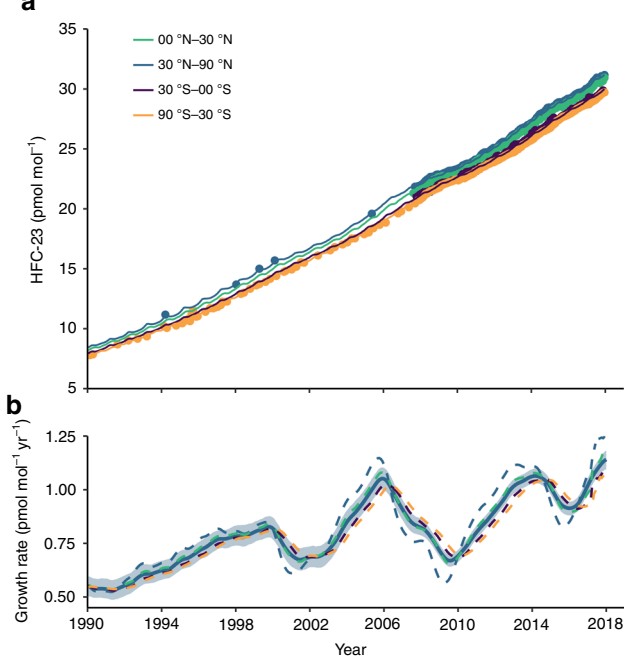

**Fig. 1 Global HFC-23 emissions and HCFC-22 production. a** Top-down global HFC-23 emissions (blue line) and uncertainties (blue shaded area; $1\sigma$, incorporating uncertainties due to the prior constraints, measurements, model representations of the data, calibration scale and HFC-23 lifetime, see Methods section) derived from Advanced Global Atmospheric Gases Experiment (AGAGE) data and the 12-box model (inferred emissions available in Supplementary Table 4). All other lines represent global (dashed) or sub-global (dotted) bottom-up estimates: developing countries emissions estimates (HCFC-22 production multiplied by an emissions factor (EF)) are shown in dark green; developed countries emissions obtained from United Nations Framework Convention on Climate Change (UNFCCC) reports are shown in black; the developing countries no abatement global total emissions estimates (sum of emissions from the unabated developing countries estimates and developed countries reports) are shown in orange; the developing countries with abatement global estimates are shown in green (equivalent to the developing countries no abatement estimates but with clean development mechanism (CDM) and post-2015 reports subtracted); the red dotted line is equivalent to the developing countries no abatement estimates, but with the maximum reported CDM abatement amount subtracted each year after 2011 (9.0 Gg yr$^{-1}$), thus estimating the maximum possible legacy of the CDM. The hatched and cross-hatched areas represent the CDM and post-2015 abatement (Chinese and Indian reported emissions reductions combined), respectively. **b** Total HCFC-22 production (dashed orange line) from developing countries (dark green dotted line) and developed countries (black dotted line). All data points are shown as the mid-point of the respective year.

five core Advanced Global Atmospheric Gases Experiment (AGAGE) stations[17,18], located in remote sampling locations at Mace Head (Ireland), Trinidad Head (California, USA), Ragged Point (Barbados), Cape Matatula (American Samoa) and Cape Grim (Tasmania, Australia), in conjunction with the AGAGE 2-D atmospheric 12-box model and a Bayesian inversion methodology (see Methods section)[6]. Our estimates are consistent with the most recent prior study, which covered the period until 2015[6], showing an overall increase in HFC-23 emissions over the

**Fig. 2 HFC-23 observations, model-derived mole fractions and growth rates. a** Modelled atmospheric HFC-23 mole fractions in pmol mol$^{-1}$ for the four equal-mass latitudinal subdivisions in the 12-box model based on in situ measurements at the core background measurement sites (points), northern hemispheric flask samples (blue circles; only shown prior to 2007) and Cape Grim Air Archive data (orange circle, only shown prior to 2007). **b** Model-derived annual HFC-23 growth rates (global - blue solid line with 1-sigma uncertainty indicated by shading; dashed lines show semi-hemispheric growth rates) in pmol mol$^{-1}$ yr$^{-1}$.

previous three decades, but with a substantial decline during the CDM period and a small drop between 2014 and 2015. The AGAGE data show a renewed increase in the HFC-23 growth rate from 2016, which reached $1.1 \pm 0.05$ pmol mol$^{-1}$ yr$^{-1}$ in 2018, when global annual mean mole fractions were 31.1 parts per trillion (pmol mol$^{-1}$; Fig. 2). In contrast, a forward model run using our developing country with abatement estimate suggests that the global mean mole fraction growth rate should have declined to less than zero after around 2016 (Supplementary Fig. 1). Our top-down emissions estimate shows that the observed growth has been driven by an increase in HFC-23 emissions from 2016 to a new maximum of $15.9 \pm 0.9$ Gg yr$^{-1}$ in 2018 (Fig. 1a), despite the aforementioned reported emission reductions post-2015.

**Discrepancies between top-down and bottom-up estimates.** In common with previous studies, our bottom-up HFC-23 estimate, based on HCFC-22 production and UNFCCC reports, is in good agreement with emissions inferred from atmospheric observations prior to the CDM period (2006)[6,8]. During the CDM period, the measurement-derived emissions show a decline to a minimum in 2009, as expected from CDM reports. Between 2009 and 2012, both top-down and bottom up (with developing country abatement) HFC-23 emissions estimates increased and were in good agreement, within the uncertainty of the top-down estimate.

After 2012, our top-down estimates grow more slowly than would be expected from HCFC-22 production and the decline in reported CDM abatement (Fig. 1a); CDM abatement declines to zero by the end of 2014, resulting in our abatement and no-abatement bottom-up estimates nearly converging at 20.8 Gg yr$^{-1}$, compared to the top-down estimate of 14.5 Gg yr$^{-1}$.

This corresponds to a maximum discrepancy between top-down and bottom-up with abatement of 6.3 Gg yr$^{-1}$ in 2014. If the full abatement capacity installed during the CDM had continued, our bottom-up estimate would be substantially lower in 2014; if we assume that the maximum amount abated during the CDM (9.0 Gg yr$^{-1}$ in 2011) had continued (red dotted line in Fig. 1a), our bottom-up estimate would be around 11.8 Gg yr$^{-1}$ in 2014. Based on these considerations, we suggest that the growth in top-down emissions between 2012 and 2014 can be explained by new emissions from newly installed, at least partly unabated, HCFC-22 production capacity, combined with the switching off of some, but not all, abatement that was installed during the CDM period (consistent with observations from one previous study in India[13]).

A previous study suggested that the decline in HFC-23 emissions between 2014 and 2016 was consistent with the timing of the reductions undertaken by China and reported under its HPPMP[6]. However, here we show that the magnitude of this decline in top-down emissions between 2014 and 2016 (around 1.6 ± 1.6 Gg (2σ uncertainty), or 11%) is substantially smaller than the anticipated emission reductions (around 13 Gg from China, which would represent around 72.4% of the 2014 top-down value, Fig. 3). We note that this period coincided with a drop in HCFC-22 production of 6.3%, and therefore, we propose that HFC-23 emissions may have largely followed this trend (Fig. 1a). Indeed, when HCFC-22 production increased substantially in 2017, global top-down HFC-23 emissions also rebounded. In contrast, when reported reductions from China and India are considered together with developed country reports to the UNFCCC (hatched area in Fig. 1a), our bottom-up estimate of global total HFC-23 emissions for 2017, 2.4 Gg yr$^{-1}$, is 12.5 ± 0.7 Gg yr$^{-1}$ lower than our top-down value. Global emissions continued to grow in 2018. Therefore, it seems likely that the HFC-23 emission reductions reported since 2015 have not been successfully implemented until at least the beginning of 2019, or there is substantial, unreported production of HCFC-22 from which HFC-23 is vented, or some combination thereof.

## Discussion

Given the magnitude of the mismatch between our top-down values and the expected developing countries with abatement HFC-23 emissions in 2017, it is likely that China has not been successful in meeting its target of abating 98% of HFC-23 emissions; this target would have resulted in 10.5 Gg yr$^{-1}$ being abated in 2017, whereas our inferred total developing world abatement is only 4.7 Gg yr$^{-1}$ (Fig. 3). Alternatively, or perhaps in combination, we cannot exclude the possibility that there is substantial, unreported production of HCFC-22 from which HFC-23 is vented. If the Chinese HFC-23 emissions reductions have in fact taken place, there would need to have been a coincident increase in emissions of 780% from developed countries, or 690% from India between 2015 and 2017, or unreported HCFC-22 production (with no HFC-23 abatement) of around 4250% of the reported global total in 2017. Given that India's pledged abatement (1.9 Gg yr$^{-1}$ in 2017) is substantially smaller than China's, and smaller than our inferred developing world-total abatement, it is not clear from the top-down results whether India's abatement measures have been successful (Fig. 3). Notably, our estimate of total developing world abatement is lower between 2015 and 2017 than at any point since 2007.

The integrated difference between our inferred top-down emissions and the bottom-up estimate that considers reported emission reductions, is 24.4 Gg between 2015 and 2017 (the area between the dashed green and solid blue lines in Fig. 1a). Therefore, our results imply a missed opportunity to avoid the

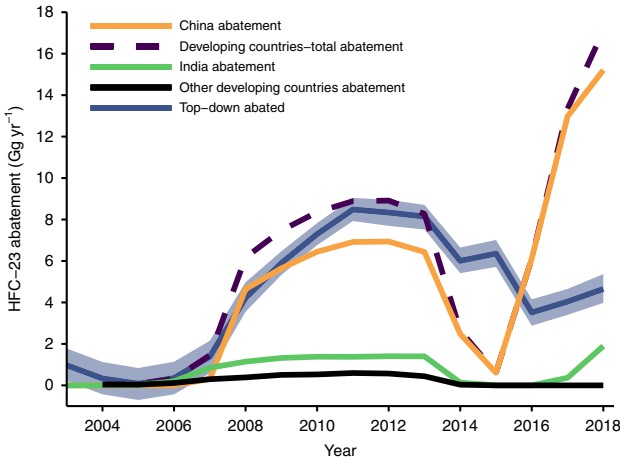

**Fig. 3 Reported abatement versus estimates based on atmospheric data.** Total developing countries HFC-23 abatement (blue line, with blue shaded area showing 1σ uncertainty) calculated from the developing country no abatement bottom-up emissions estimates minus the top-down emissions derived from the 12-box model with developed world reported emissions removed. Chinese (orange line) and Indian (green line) abatement are compiled from the United Nations Framework Convention on Climate Change's Clean Development Mechanism (CDM) reports (2010–2014), Chinese hydrochlorofluorocarbons production phase-out management plan (HPPMP) reported emission reductions, and the Indian executive order to abate all HFC-23 emissions from 13 October 2016. Other developing country abatement as part of the CDM is shown in the black line. Developing countries-total abatement (dashed purple line) shows the summation of all developing country abatement as part of the CDM, Chinese HPPMP and Indian executive order.

equivalent of around 309 Tg $CO_2$ emissions during this period. These extra emissions are roughly equivalent to the total greenhouse gas emissions of Spain in 2017 (which, including land use, land use change and forestry, were 302 Tg $CO_2$-equivalent)[19]. Therefore, rapid implementation of these abatement plans would have a substantial beneficial influence on climate.

## Methods

**Atmospheric measurements**. Atmospheric measurements were taken from the five longest-running baseline stations in the AGAGE network (Supplementary Fig. 2), including Mace Head (Ireland; 53.3°N, 9.9°W), Trinidad Head (California, USA; 41.0°N, 124.1°W), Ragged Point (Barbados; 13.2°N, 59.4°W), Cape Matatula (American Samoa; 14.2°S, 170.6°W) and Cape Grim (Tasmania, Australia; 40.7°S, 144.7°E)[17]. Together, these observatories provide long-term in situ measurements of a wide range of greenhouse gases and ozone-depleting substances. At each site, HFC-23 ($CHF_3$) is detected using the Medusa Gas Chromatography Mass Spectrometry (GCMS) analytical system, which is capable of making up to 12 calibrated ambient measurements per day. The routine operation of the Medusa GCMS has been described previously[20,21]. HFC-23 is reported relative to the SIO-07 calibration scale, which links remote ambient measurements to a set of gravimetrically prepared 'primary' standards via a hierarchy of compressed gas standards maintained in 34 L electro-polished stainless-steel canisters (Essex Industries, Missouri, USA). Analyses of the working 'reference' standard bracket each ambient measurement, and this gas is compared to a 'tertiary' gas on a weekly basis. Daily reference gas measurements have a precision of ~0.5–1%. System blanks and laboratory air analyses are also conducted weekly, to quantify possible influences from carrier gas impurities and system leaks. In situ AGAGE measurements of HFC-23 have been available since 2007. Prior to 2007, measurements were obtained from the Cape Grim Air Archive[8,22,23] and a suite of air samples collected in the Northern Hemipherer[24,25]. Air archive samples were analysed at the Commonwealth Scientific and Industrial Research Organisation Marine and Atmospheric Research (Aspendale, Australia) and Scripps Institute of Oceanography (La Jolla, California) using the same medusa GCMS analytical system as in situ measurements[6].

**AGAGE 12-box model and inverse method**. Global HFC-23 emissions were estimated using the in situ and archive data and a two-dimensional atmospheric

chemistry and transport model, the AGAGE 12-box model[27,28,31]. The AGAGE 12-box model splits the atmosphere into four equal mass latitude bands (divisions at 30–90°N, 0–30°N, 0–30°S and 30–90°S), and is split vertically at 200, 500 and 1000 hPa. The lifetime of HFC-23 was ~240 years in the model, based on the HFC-23—hydroxyl reaction rate estimated by Burkholder et al.[29] and stratospheric lifetime from Ko et al.[4]. The model uses annually repeating meteorology and a hydroxyl radical concentration climatology[30], tuned to match the growth rate of methyl chloroform[31]. Emissions were inferred using a Bayesian method in which the emissions growth rate was weakly constrained a priori[26,27]; a prior emissions growth rate of zero plus or minus 20% of the maximum bottom-up emissions from Miller et al.[8] was assumed (i.e. a priori emissions growth of 0 ± 2.7 Gg yr$^{-1}$). In the inversion, uncertainties on the measurements were assumed to be equal to the monthly baseline variability for the high-frequency in situ data (thereby incorporating measurement repeatability and a model representation error term related to the sub-monthly timescales not resolved by the model). Uncertainties on the archived data points were assumed equal to the quadratic sum of the repeatability and the mean variability during the in situ data period, scaled by the mole fraction difference between the archive and in situ data (also an approximation of the model representation error for those archive samples). Following the Bayesian method in Rigby et al.[26], the posterior uncertainties are dependent on these assumed prior and measurement uncertainties, and, following Rigby, et al.[27], are augmented with a term related to systematic uncertainties in the calibration scale (~3%) and loss frequency (~21%[4]). The uncertainty in the derived global, annual emissions introduced by using inter-annually repeating meteorology could not be assessed. However, previous comparisons of box model and three-dimensional top-down emissions estimates for other species suggests that this source of uncertainty will be substantially smaller than the magnitude of the emissions changes that are the subject of this paper[32]. The AGAGE 12-box model HFC-23 emission estimates are in Fig. 2 and model/measurement comparison data is shown in Supplementary Fig. 3.

**Bottom-up emissions estimates.** HFC-23 is produced as a by-product during the manufacturing process of HCFC-22 (chlorodifluoromethane, $CHClF_2$). HCFC-22 production and consumption is controlled under the Montreal Protocol for both developed countries (non-Article 5 under the Montreal Protocol and Annex I under the UNFCCC, including Turkey and excluding Israel) and developing countries (Article 5 and non-Annex I, including Israel). As such, HCFC-22 production data for dispersive and feedstock applications is reported by each party to the convention. Total HCFC-22 production data for dispersive and non-dispersive uses was obtained from the UNEP HCFC database. However, country-specific data is considered confidential and only aggregated values for developing and developed countries are shown (see Supplementary Table 2). A bottom-up HFC-23 emissions inventory was compiled using data from the 2019 Common Reporting Format of NIR submitted to the UNFCCC[12] for developed countries (see Supplementary Table 1). As developing (Article 5) countries are not required to submit bottom-up emission estimates to the UNFCCC, HCFC-22 production data and time-varying emission factors (EFs) were used to generate HFC-23 emission estimates. Historically, the total mass of HFC-23 emitted was assumed to be equivalent to 4% of total HCFC-22 production, declining to 3% in the last decade[6]. From 2012, EFs used were derived from data reported by the Executive Committee of the Multilateral Fund[9–11]. Total HFC-23 emissions, shown in Fig. 1a as bottom-up [developing countries no abatement], are the summation of the developed and developing countries' totals (see Supplementary Table 1).

Beginning in 2003, selected manufacturers of HCFC-22 were eligible to produce CER credits for the destruction of HFC-23. In total, 19 facilities (11 in China, 5 in India and 1 in Mexico, South Korea and Argentina, respectively) were registered for funding under the CDM. An approved baseline methodology, AM0001, was issued to ensure consistent reporting of key parameters, including the quantity of HFC-23 (measured via inline GC) upstream and downstream of the abatement system, relative to the total quantity of HCFC-22 produced. The total mass of HFC-23 that was sold, vented or stored was also monitored to ensure that CERs were only issued for eligible quantities of destroyed HFC-23. During the CDM period (2003–2014), HCFC-22 manufacturers published a combined 474 monitoring reports (typically spanning 2–6 months of production; https://cdm.unfccc.int/Projects/registered.html). For each facility, the total annual mass of destroyed HFC-23 was estimated, based on the assumption that the rate of HCFC-22 production (and therefore production and destruction of HFC-23) remained constant over the course of any given monitoring period (for some facilities, monitoring periods spanned multiple years). These totals were then combined to estimate the total global destruction of HFC-23 (53.3 Gg, Fig. 1a, hatched region). Compiled CDM data is available in Supplementary Table 3.

Post-CDM, a number of developing countries pledged to reduce their HFC-23 emissions. The government of the People's Republic of China reported reductions in the emissions of HFC-23 by-product generated during HCFC-22 production between 2015 and 2017 (inclusive) as part of the HPPMP[10]. An executive order issued by the Indian government on 13 October 2016 requires producers of HCFC-22 to destroy HFC-23 by-products from thereafter[9,14]. Prior to the executive order, Indian HCFC-22 plants were shown to be venting HFC-23 to the atmosphere[13]. Both of the commitments to reducing HFC-23 emissions are included in the

bottom-up inventory, shown in Fig. 1a as bottom-up [developing countries with abatement]. Compiled HFC-23 data are available in Supplementary Table 1.

## Data availability

Atmospheric measurement data from the AGAGE network is available at http://agage.mit.edu/data. HCFC-22 inventory data is available from the Ozone Secretariat (https://ozone.unep.org/) and Annex I National Inventory Reports to the United Nations Framework Convention on Climate Change are available from https://unfccc.int/process-and-meetings/transparency-and-reporting/reporting-and-review-under-the-convention/greenhouse-gas-inventories-annex-i-parties/national-inventory-submissions-2019. Data used from the Multilateral Fund are text cited within the text.

## Code availability

AGAGE 12-box model code will be made available upon request by contacting Matt Rigby.

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

## Acknowledgements

The authors particularly thank the continued cooperation and efforts of the station operators, who oversee the day-to-day running of the AGAGE sites. We specifically thank the NASA Upper Atmosphere Research Program for its continuing support of AGAGE, including its 5 core stations, through grants NNX16AC98G to MIT and NNX16AC96G and NNX16AC97G to SIO. Observations at Mace Head, Ireland, are partially supported by NASA and by the Department for Business, Energy & Industrial Strategy (BEIS, UK, formerly the Department for Energy and Climate Change), contracts 1028/06/2015 and 1537/06/2018 to the University of Bristol. Ragged Point, Barbados, is partially supported by NASA, and by the National Oceanic and Atmospheric Administration (NOAA, USA) through contract RA-133-R15-CN-0008 to the University of Bristol. Observations at Cape Grim, Australia, are supported largely by the Australian Bureau of Meteorology, CSIRO, the Australian Department of the Environment and Energy (DoEE), NASA and Refrigerant Reclaim Australia (RRA). M.R. is supported by Natural Environment Research Council grant NE/I021365/1. In addition, we are grateful for contributions from the Multilateral Fund Secretariat and the Ozone Secretariat.

## Author contributions

P.B.K., J.M., S.J.O., D.Y., P.G.S., K.M.S., P.J.F., C.M.H., P.K.S., R.G.P. and R.F.W. contributed observational data. K.M.S. and D.S. compiled the bottom-up inventory. K.M.S., D.S. and M.R. carried out atmospheric model simulations and wrote the paper, with contributions from all co-authors.

## Competing interests

The authors declare no competing interests.
