## [Peer Review File · Nature Communications]

Reviewers' comments:

Reviewer #1 (Remarks to the Author):

Title: Increase in global emissions of HFC-23 despite near-total expected reductions
- by Stanley et al.

General comments

The Kigali Amendment (KA) to the Montreal Protocol on Substances that Deplete the Ozone Layer entered into force on 1 January 2019, following ratification by 65 countries. HCFC-22 is a major source of HFC-23 emissions, which is a strong greenhouse gas with GWP100 of 12690 times that of CO₂. The Chinese production capacity of HCFC-22 accounts for 66% of the global HCFC-22 production (593,047 mt out of 895,158 mt). HFC-23 emissions from HCFC-22 production were controlled in most developing countries (viz. India, Mexico, South Korea, Argentina) due to the Clean Development Mechanism (CDM) of the Kyoto Protocol, except China where 36% of HCFC-22 production was controlled by CDM (see: <http://xbna.pku.edu.cn/CN/Y2012/V48/I2/310>). It may be noted that HCFC-22 production in North Korea and Venezuela is not controlled – even during the CDM regime. In 2017, HCFC-22 production in these two countries were 451 and 273 Mt respectively.

In 2015, the Chinese National Development and Reform Commission (NDRC) announced that it plans to achieve abatement of all HFC-23 emissions by 2019. This would imply installing destruction technology in all plants currently not covered by CDM and ensuring that the destruction technology on plants covered under CDM is being operated and maintained (see: <https://www.atmos-chem-phys.net/17/2795/2017/>). China reported a 98% reduction of HCFC-22 production-related HFC-23 emissions by 2017. In its Intended Nationally Determined Contribution (INDC) submitted in June 2015, China reiterated its commitment under the Montreal Protocol to achieve effective control on emissions of HFC-23 by 2020. In the 28th Meeting of the Parties to the Montreal Protocol in October 2016 in Kigali, the Indian government presented domestic legislation that mandates control of HFC-23 emissions.

On this background, this study assesses that HFC-23 abatement measures in HCFC-22 production facilities should reduce global HFC-23 emissions by 87% between 2014 and 2017. However, the atmospheric observations show that emissions have increased and in 2018 were higher than at any point in history. This observation is critically important to validate the successful implementation of HFC-23 emission control as declared by the parties to the Montreal Protocol (particularly China and India). Additionally, tight scrutiny of the global HFC-23 emissions will be useful to check the unreported HCFC-22 production.

Without questioning the novelty of such an interesting work done by the authors - I can see some of these studies in the reference section, however, it will be useful for the target audience of the journal if the authors clearly mention what is new in this study as compared to the available literature (e.g. Simmonds et al., 2018) in the introductory section. I can see that some of the authors also contributed to Simmonds et al. (2018). Below are a few suggestions which the author may find useful for gaining the clarity and bringing the transparency in work.

1) Page 1, line 3: ...(75% in 2017)... : In 2017, total HCFC-22 production of China and India was 657,556 metric ton – approximately 73% of the global HCFC-22 production of 895,459 metric ton (see: <http://multilateralfund.org/82/English/1/8268.pdf>). The contribution of all Article 5 countries was approximately 75% in 2017.

2) Page 1, line 17: Please correct – "...have recently have been..." or "...have recently been...?"

3) Page 1, line 29-32: As I remember most of the CDM developers presented emissions in terms of CO₂ equivalent (CER) using IPCC/SAR GWP100 whereas this study uses the recent 2018 SAP values. Just for the clarity which GWP100 values were used to calculate HFC-23 reductions from CDM projects (Table S3) or the detailed PDDs also provide HFC-23 emissions separately?

4) Page 1, line 29-32: Not all Article 5 countries are non-Annex I parties to the UNFCCC – for e.g. Turkey. Similarly, not all non-Article 5 countries are Annex-I parties to the UNFCCC – for e.g. Israel.

5) Page 2, line 41-42: You have already explained HFC (Hydrofluorocarbon) and HCFC (Hydrochlorofluorocarbon) above (page 1). Instead you may use preferred IUPAC name for HCFC-22 (Chlorodifluoromethane) and HFC-23 (Trifluoromethane).

6) Page 2, line 57-58: This information is already provided above (line 39-40).

7) Page 4, Figure 2: How about showing the HFC-23 abatement below the horizontal axis?

8) Page 4, Figure 2: The Clean Development Mechanism (CDM) is one of the Flexible Mechanisms defined in the Kyoto Protocol that provides for emissions reduction projects which generate Certified Emission Reduction units (CERs) which may be traded in emissions trading schemes. In the title please note that CDM belongs to the United Nations Framework Convention on Climate Change not the United Nations Environment Programme.

9) Page 7, line 223: As per journal citation style Miller et al. [15] not Miller et al., 2010.

10) Page 7, line 227: As per journal citation style Burkholder et al. [26] not Burkholder et al., 2015 [26].

11) Page 8, line 232-234: Not all Article 5 countries are non-Annex I parties and not all non-Article 5 countries are Annex-I parties. You may simply say Article 5 (mostly non-Annex-I developing countries) and non-Article 5 (mostly annex-I developed countries) or vice-versa.

12) Page 8, line 258-264: Information already provided in page 2.

Reviewer #2 (Remarks to the Author):

This is a very important, and concerning, study reporting renewed large emissions of the potent greenhouse gas HFC-23, despite apparent reported actions to dramatically reduce a significant global quantity of these emissions. As the authors report, it seems likely that the HFC-23 emission reductions reported from China and India since 2015 have not been successfully implemented. This study has important consequences for climate protection and for the Kigali Amendment to the Montreal Protocol. The manuscript is written clearly, is based on methodical and high quality global measurements and modelling, and the conclusions are well justified based on the results. I recommend publication, after considering some suggestions for improvement.

Fig 1. The bottom up emissions 1 (and the “top-down abated” emissions in Fig 2) depend to a large extent on the emission factor to convert from HCFC-22 production (for developing countries). While the EF obviously gives good agreement compared to the top-down emissions prior to the CDM period/ 2006, is there any evidence that this emission factor, i.e. the manufacturing process, has likely not changed over time?

Ln 88. Can the authors comments on why was there was no reported CDM abatement of HFC-23 emissions at all by 2014/2015? Also, the “developing country with abatement” scenario (Fig 1 green line) massively exceeds the top down HFC-23 emissions over this period, suggesting that abatement did actually occur.

Minor comments

Title. What is a “near-total reduction”? “Near-total removal” makes sense, but not reduction.

Ln 17. "have recently have"

Fig 1 legend. Typo "pot-2015".

Reviewer #3 (Remarks to the Author):

Overview

The manuscript presents a bottom-up annual emissions inventory for HFC-23, an unwanted by-product of the production of HCFC-22 and powerful greenhouse gas. The authors have assembled reported production rates of HCFC-22 (monitored under the Montréal Protocol) and the amount of HFC-23 typically generated during production in order to estimate an upper-limit for HFC-23 production in time. The authors then separately estimate what emissions of HFC-23 would have been if HFC-23 abatements reported by developing countries would have prevented its release into the atmosphere. The authors then compare these emissions to the time series of emissions implied by measurements of HFC-23 in surface air sampled at five sites using an inverse model. The top-down emissions (which imply global emissions continue to increase in time) to be inconsistent with self-reported massive decreases in HFC-23 emissions reported by developing nations, especially China.

The results are extremely provocative and important for climate change mitigation if true, and I generally support publication. However, I would like the authors to provide more information about their methods and uncertainties, as outlined below, especially in light of the geopolitical implications of the conclusions.

Broad Comments

I do not disbelieve the conclusions based on my scientific experience. However, I think the manuscript needs to provide more information in order to strongly support its conclusions. One particular weakness is that the text focuses on the large drop in emissions predicted by the Chinese commitments beginning in 2015, a period in which the top-down constrained emissions predict a modest increase. However, the period from 2012 to 2014 is ignored, in which the top-down constraint implies emissions were much lower than the bottom-up inventory. How do we know that the representative and transport errors of a discretely sampled gas are not simply smoothing the inferred emission changes in the inverse model? Why did the top-down and bottom-up methods decouple in 2012, before the self-reported large reductions in China and India begin? How is the 1sigma bars exactly determined and what does it encapsulate? In particular, if the model reuses the same meteorology each year — as per the Methods section — how does real-world interannual variability in transport (such as might be expected with the transition from a La Niña state to strong El Niño state during this period) influence the results? I'm sure the authors have thought about all these and have adequate answers, but think that some discussion of this needs to be brought into the main text, or at least the supplement before being suitable for publication.

As a lesser point, I find it unusual for a top-down evaluation study that there are no observations shown in the main text. I would move the present Figure 3 into the supplement (the five locations are already described in the text with words), and a version of Figure S1 into the main text.

Minor Comments

L19 and L226 - Why does the manuscript use two different HFC-23 lifetimes?

Fig. 1 caption - "pot-2015" should be "post-2015"

I find Figure 1 overly confusing for the information it is trying to convey (although, this is personal preference). Fig. 1a includes the red-dotted line that is never discussed in the text anywhere aside

from the caption, and its purpose for being shown is not made clear; I think it could be cut. I understand the purpose of the hatching but personally think it would be better conveyed using simple vertical lines and maybe rectangular background shading to delineate the temporal periods of interest (pre-CDM abatement vs. CDM period vs. post-2015 reductions). The thing that matters most here is the comparison of the top-down emissions (blue line) vs. the reported emissions (green line). To that end, the dots showing the developed vs. developing nations non-abatement inventories seem superfluous and might be best relegated to the Supplement. Fig. 1b would convey the same information more simply as a stacked area chart.

Fig. S1 caption – dashed lines presumably show latitude band growth rates, not hemispheric growth rates.

We thank the reviewers for their thoughtful comments. Below, we respond to each of the comments in turn, with our responses are in red.

Reviewer #1 (Remarks to the Author):

Title: Increase in global emissions of HFC-23 despite near-total expected reductions
- by Stanley et al.

General comments

Without questioning the novelty of such an interesting work done by the authors - I can see some of these studies in the reference section, however, it will be useful for the target audience of the journal if the authors clearly mention what is new in this study as compared to the available literature (e.g. Simmonds et al., 2018) in the introductory section.

To clarify what is new about the study, and abide by Nature Communications manuscript guidelines, two paragraphs have been added to the end of the introduction (one of which has been moved up from the results section), which state previous knowledge of HFC-23 emissions and outline the new information presented in this study. The two paragraphs (lines 20 – 35, page 2, in the revised manuscript) added are as follows:

“Previous studies, based on in situ atmospheric observations and firm air measurements, have shown an increase in the global annual mean mole fraction of HFC-23 from near zero in early 1960 to 28.9 ± 0.6 pmol mol⁻¹ by the end of 2016[1, 5, 6]. These data, when combined with a model of atmospheric transport and chemistry can be used to infer global emissions. Such “top-down” methods have previously shown that global HFC-23 emissions grew from 4.2 ± 0.7 Gg yr⁻¹ in 1980 to 13.3 ± 0.8 Gg yr⁻¹ in 2006, before declining rapidly to 9.6 ± 0.6 Gg yr⁻¹ in 2009 in response to emission reductions from developed countries and as a result of the UNFCCC Clean Development Mechanism (CDM)[6, 8]. As the CDM period ended, HFC-23 emissions grew to 14.5 ± 0.6 Gg yr⁻¹ in 2014, before declining again, to 12.7 ± 0.6 Gg yr⁻¹, in 2016[6].”

Here, we present an update of global HFC-23 emissions derived from atmospheric observations, based on new data from 2015 to the end of 2018. In addition, we compile a new inventory-based HFC-23 emissions estimate through to the end of 2017 that includes reported emission reductions by China and India. We find that in 2018, observation-based HFC-23 emissions are higher than at any point in history (15.9 ± 0.9 Gg yr⁻¹), whilst inventory-based emissions are at the lowest in the past 17 years (2.4 ± 0.9 Gg yr⁻¹ in 2017) when reported emission reductions were included. Due to the magnitude of the discrepancy between reported emissions reductions and emissions inferred from the atmospheric data, it is highly likely that developing countries have been unsuccessful in meeting their reported emissions reductions. Alternatively, or additionally, there may be substantial unreported production of HCFC-22 at unknown locations resulting in unaccounted-for HFC-23 by-product being vented to the atmosphere.”

1) Page 1, line 3: ...(75% in 2017)... : In 2017, total HCFC-22 production of China and India was 657,556 metric ton – approximately 73% of the global HCFC-22 production of 895,459 metric ton (see: <http://multilateralfund.org/82/English/1/8268.pdf>). The contribution of all Article 5 countries was approximately 75% in 2017.

A revision to these numbers was made in Multilateral Fund (MLF) documents (see: <http://multilateralfund.org/82/English/1/8268c1.pdf> for the Corrigendum), where Chinese production was increased to 644,721 (from 593,047) metric tonnes and the resultant total for all countries increased to 947,132 (from 895,459) metric tonnes. With the revisions from the Corrigendum, HCFC-22 production from China and India (64,509 metric tonnes) equates to 75% of global emissions.

2) Page 1, line 17: Please correct – “...have recently have been...” or “...have recently been...”?
Many thanks for showing us the error, which has been corrected in the revised manuscript (Page 2, line 12).

3) Page 1, line 29-32: As I remember most of the CDM developers presented emissions in terms of CO₂ equivalent (CER) using IPCC/SAR GWP100 whereas this study uses the recent 2018 SAP values. Just for the clarity which GWP100 values were used to calculate HFC-23 reductions from CDM projects (Table S3) or the detailed PDDs also provide HFC-23 emissions separately?

The individual CDM reports that were used to compile each non-Article 5 country's yearly abatement report destroyed HFC-23 in metric tonnes (MT), not in terms of CO₂-equivalent, hence no GWP100 value was used.

4) Page 1, line 29-32: Not all Article 5 countries are non-Annex I parties to the UNFCCC – for e.g. Turkey. Similarly, not all non-Article 5 countries are Annex-I parties to the UNFCCC – for e.g. Israel.

We based our definition of Annex I parties to the UNFCCC on those who submit a National Inventory Report and CRF documents to the Protocol secretariat, which includes Turkey but excludes Israel. Non-Article 5 countries were based on the I/12E decision of the 2018 Handbook to the Montreal Protocol (see: https://ozone.unep.org/sites/default/files/MP_handbook-english-2018.pdf, p.321), which excludes both Turkey and Israel. However, as Israel does not submit a National Inventory Report to the UNFCCC, it was included within this definition. To make the manuscript clearer, we have altered lines 41 – 45 (Page 2) in the revised manuscript to state:

“We compiled a “Developing Country No Abatement” bottom-up HFC-23 emissions estimate (Supplement Table S1) based on HCFC-22 production data obtained from the UNEP HCFC database[9] (Supplement Table S2) and available time-varying emissions factors[6, 10–12] for developing nations (defined here as Article 5 countries under the Montreal Protocol and non-Annex I under the UNFCCC, including Israel). For developed countries (non-Article 5 countries under the Montreal Protocol and Annex I under the UNFCCC, including Turkey but excluding Israel)”...

The Table caption in Supplement Table S1 was also changed to make the definition of developed and developing country clearer and now reads:

“Table S1. “Bottom-up” HFC-23 emissions to the atmosphere (Gg yr⁻¹). Developed (Non-Article 5 under the Montreal Protocol; Annex I under the United Nations Framework Convention on Climate Change; UNFCCC; including Turkey but excluding Israel) countries data was compiled and aggregated from the 2019 National Inventory Reports submitted to the UNFCCC by each Annex I party[3]. Developing (Article-5 under the Montreal Protocol; non-Annex I under the UNFCCC; including Israel) countries’ data”...

5) Page 2, line 41-42: You have already explained HFC (Hydrofluorocarbon) and HCFC (Hydrochlorofluorocarbon) above (page 1). Instead you may use preferred IUPAC name for HCFC-22 (Chlorodifluoromethane) and HFC-23 (Trifluoromethane).

We thank Reviewer 1 for pointing out this second inclusion of the definition for HFCs and HCFCs. The explanation of the names was included in the text as it refers to the name of the Indian executive order that requires all HCFC-22 producers to destroy by-product HFC-23 emissions. As it is the name of an executive order, we will leave it in the manuscript as it is. However, if the editor would like us to incorporate this recommended change, we will do so willingly.

6) Page 2, line 57-58: This information is already provided above (line 39-40).

Many thanks for the suggestion, this sentence has been removed from the paragraph (lines 69 – 78, page 3 of the revised manuscript).

7) Page 4, Figure 2: How about showing the HFC-23 abatement below the horizontal axis?

We thank the reviewer for their suggestions; however, we feel that the current layout of HFC-23 abatement shown on the y-axis and time on the x-axis allows for easier comparison with Figure 1 for the reader. In the revised manuscript, Figure 2 has now become Figure 3 (page 6).

8) Page 4, Figure 2: The Clean Development Mechanism (CDM) is one of the Flexible Mechanisms defined in the Kyoto Protocol that provides for emissions reduction projects which generate Certified Emission Reduction units (CERs) which may be traded in emissions trading schemes. In the title please note that CDM belongs to the United Nations Framework Convention on Climate Change not the United Nations Environment Programme.

Many thanks for pointing out our error. We have corrected the Figure caption (now Figure 3, page 6, in the revised manuscript).

9) Page 7, line 223: As per journal citation style Miller et al. [15] not Miller et al., 2010.

Many thanks for pointing out our error, the 2010 was removed from the in-text citation (page 7, line 163).

10) Page 7, line 227: As per journal citation style Burkholder et al. [26] not Burkholder et al., 2015 [26].

Again, many thanks for pointing out our error, the 2015 was removed from the in-text citation (page 7, line 167).

11) Page 8, line 232-234: Not all Article 5 countries are non-Annex I parties and not all non-Article 5 countries are Annex-I parties. You may simply say Article 5 (mostly non-Annex-I developing countries) and non-Article 5 (mostly annex-I developed countries) or vice-versa.

As for Reviewer 1 point 4, we thank the reviewer for their comment. Our definition for Annex I parties to the UNFCCC is based on those who submit a National Inventory Report and CRF documents to the Protocol secretariat, which includes Turkey but excludes Israel, as HFC-23 data was derived for these parties from the CRF documents. Non-Article 5 countries were based on the I/12E decision of the 2018 Handbook to the Montreal Protocol (see: https://ozone.unep.org/sites/default/files/MP_handbook-english-2018.pdf, p.321), which excludes both Turkey and Israel. However, as Israel does not submit a National Inventory Report to the UNFCCC, it was included within this definition. To make the manuscript clearer (on lines 175 to 178, page 7, in the revised manuscript), we have altered the sentence to state:

“HCFC-22 production and consumption is controlled under the Montreal Protocol for both developed countries (non-Article 5 under the Montreal Protocol and Annex I under the UNFCCC, including Turkey but excluding Israel) and developing countries (Article 5 and non-Annex I, including Israel).”

12) Page 8, line 258-264: Information already provided in page 2.

The sentence (on page 8, lines 201 to 204 of the revised manuscript) has been reworded to remove the repetition from the previous sections to read:

“The government of the People’s Republic of China reported reductions in the emissions of HFC-23 by-product generated during HCFC-22 production between 2015 and 2017 (inclusive) as part of the HPPMP[11]. An executive order issued by the Indian government on 13th October 2016 requires producers of HCFC-22 to destroy HFC-23 by-products from thereafter [10, 15].”

Reviewer #2 (Remarks to the Author):

Fig 1. The bottom up emissions 1 (and the “top-down abated” emissions in Fig 2) depend to a large extent on the emission factor to convert from HCFC-22 production (for developing countries). While the EF obviously gives good agreement compared to the top-down emissions prior to the CDM period/ 2006, is there any evidence that this emission factor, i.e. the manufacturing process, has likely not changed over time?

Emission Factors (EFs) used to convert HCFC-22 production by developing countries to HFC-23 emissions as part of the “bottom-up” inventory were based on the available literature, as stated in lines 240-243 of the original manuscript, which has shown that the HFC-23 production rates have indeed changed over time [1,2]. Due to improvements in operational efficiency in HCFC-22 production plants via production line optimization, countries EFs have reduced from the historic value of 4% to 3% within the last decade[1]. This decreasing trend is also shown in Figure 2-7 in the latest Ozone Assessment report[2], where both atmospheric-derived and inventory-compiled results show decreases in the HFC-23 production rate from HCFC-22 to have decreased.

When looking at more recent data, it can be seen that for China, the largest producer of HCFC-22 (68% in 2017) and therefore by-product HFC-23, we can see that the HFC-23 production rate has further decreased from 2.78 % in 2012 to 2.41% in 2017, based on verification data provided by the Chinese Government to the MLF[3-5].

To clarify the time-varying nature of the EFs used, the following sentences have been added to the results and methods sections:

Results section, lines 41 – 43, page 2, of the revised manuscript:

“We compiled a “Developing Country No Abatement” bottom-up HFC-23 emissions estimate (Supplement Table S1) based on HCFC-22 production data obtained from the UNEP HCFC database[9] (Supplement Table S2) and available time-varying emissions factors[6, 10–12] for developing nations...”

Methods section, lines 183 – 185 (pages 7 – 8) of the revised manuscript:

“As developing (Article 5) countries are not required to submit bottom-up emission estimates to the UNFCCC, HCFC-22 production data and time-varying emission factors (EFs) were used to generate HFC-23 emission estimates”.

[1] Simmonds, P. G. et al. Supplement of “Recent increases in the atmospheric growth rate and emissions of HFC-23 (CHF3) and the link to HCFC-22 (CHClF2) production”. *Atmos. chem. Phys.* 18,4153–4169 (2018)

[2] Montzka, S. A. et al. Hydrofluorocarbons (HFCs) in Scientific Assessment of Ozone Depletion: 2018. Global Ozone Research and Monitoring Project–Report No. 58 (World Meteorological Organization, Geneva, Switzerland, 2019).

[3] UNEP. Key aspects related to HFC-23 by-product control technologies. Report reference UNEP/OzL.Pro/ExCom/79/48.(2017).

[4] UNEP. Cost-effective options for controlling HFC-23 by-product emissions. Report reference UNEP/OzL.Pro/ExCom/82/68. (2018).

[5] UNEP. Corrigendum: Cost-effective options for controlling HFC-23 by-product emissions. Report reference UNEP/OzL.Pro/ExCom/82/68/Corr.1. (2018)

Ln 88. Can the authors comments on why was there was no reported CDM abatement of HFC-23 emissions at all by 2014/2015?

The European Union banned the use of HFC-23 Certified Emission Reduction (CER) credits produced in industrial processes as part of the CDM within the European Union Emissions Trading system on May 1st 2013[1] due producers increasing HCFC-22 production above demand to increase HFC-23 by-product emissions. This effectively rendered CERs valueless and led to reported CDM abatement reducing to zero by the end of 2014. As there was no economic incentive or need to report HFC-23 abatement post-CDM, it is assumed that most countries switched off the abatement technology in the HCFC-22 production plants. However, this may not be the case and some plants may have continued to abate, as possibly seen in the discrepancy between the “developing country with abatement” scenario (Fig 1a green line) and the top-down (Fig. 1a blue line) emission estimates. It isn't clear from the information available if production plants did continue to abate and the increase in by-product HFC emissions stems from new plant capacity installations after the CDM stopped giving credits to factories.

[1] Simmonds, P. G. et al. Supplement of “Recent increases in the atmospheric growth rate and emissions of HFC-23 (CHF3) and the link to HCFC-22 (CHClF2) production”. *Atmos. chem. Phys.* 18,4153–4169 (2018)

Also, the “developing country with abatement” scenario (Fig 1 green line) massively exceeds the top down HFC-23 emissions over this period, suggesting that abatement did actually occur.

The decoupling of the top-down (blue line in Fig. 1a) and the bottom-up emissions estimates with developing country abatement included (green line in Fig. 1a) between 2012 and 2014 do suggest that abatement occurred during this period. It is not clear where this abatement comes from though; it could be that some fraction of the CDM-funded abatement systems were kept running after their CDM projects finished, or that some of the new production capacity that came online between 2013 and 2015 was either partially or fully abated. For extra

clarification, the paragraph discussing the discrepancies during the 2012 to 2015 period has been altered in the revised manuscript on lines 99 – 108 and reads as follows:

“After 2012, our top-down estimates grow more slowly than would be expected from HCFC-22 production and the decline in reported CDM abatement (Fig. 1a); CDM abatement declines to zero by the end of 2014, resulting in our “abatement” and “no-abatement” bottom-up estimates nearly converging at 20.8 Gg yr⁻¹, compared to the top-down estimate of 14.5 Gg yr⁻¹. This corresponds to a maximum discrepancy between top-down and bottom-up with “abatement” of 6.3 Gg yr⁻¹ in 2014. If the full abatement capacity installed during the CDM had continued, our bottom-up estimate would be substantially lower in 2014; if we assume that the maximum amount abated during the CDM (9.0 Gg yr⁻¹ in 2011) had continued (red dotted line in Fig. 1a), our bottom-up estimate would be around 11.8 Gg yr⁻¹ in 2014. Based on these considerations, we suggest that the growth in top-down emissions between 2012 and 2014 can be explained by new emissions from newly installed, at least partly unabated, HCFC-22 production capacity, combined with the “switching off” of some, but not all, abatement that was installed during the CDM period (consistent with observations from one previous study in India [14]).”

Minor comments

Title. What is a “near-total reduction”? “Near-total removal” makes sense, but not reduction.

We thank the reviewer for pointing out the uncertainty in the title. However; we believe that the “near-total removal” does make sense as it refers to emissions. Neither of the two other reviewers disagree with the title and therefore, we would like to keep it the same. We are of course willing to change it if the editor feels it is necessary.

Ln 17. “have recently have”

Many thanks for showing us the error, which has been corrected in the revised manuscript (page 2, line 18).

Fig 1 legend. Typo “pot-2015”.

We thank the reviewer for pointing out the typo, we have corrected this in the revised manuscript (page 4).

Reviewer #3 (Remarks to the Author):

Overview

Broad Comments

I do not disbelieve the conclusions based on my scientific experience. However, I think the manuscript needs to provide more information in order to strongly support its conclusions. One particular weakness is that the text focuses on the large drop in emissions predicted by the Chinese commitments beginning in 2015, a period in which the top-down constrained emissions predict a modest increase. However, the period from 2012 to 2014 is ignored, in which the top-down constraint implies emissions were much lower than the bottom-up inventory.

We propose that the decoupling of the top-down and the bottom-up emissions estimates between 2012 and 2014 was because not all abatement was “switched off” after the CDM projects finished and/or because the new production capacity that came online between 2013 and 2015 was either partially or fully abated. Extra information has been added to the paragraph which discusses discrepancies in top-down and bottom-up emission estimates between 2012 – 2015 and is on lines 99 – 108 of the revised manuscript. The revised paragraph reads as follows:

“After 2012, our top-down estimates grow more slowly than would be expected from HCFC-22 production and the decline in reported CDM abatement (Fig. 1a); CDM abatement declines to zero by the end of 2014, resulting in our “abatement” and “no-abatement” bottom-up estimates nearly converging at 20.8 Gg yr⁻¹, compared to the top-down estimate of 14.5 Gg yr⁻¹. This corresponds to a maximum discrepancy between top-down and bottom-up with “abatement” of 6.3 Gg yr⁻¹ in 2014. If the full abatement capacity installed during the CDM had continued, our bottom-up estimate would be substantially lower in 2014; if we assume that the maximum amount abated during the CDM (9.0 Gg yr⁻¹ in 2011) had continued (red dotted line in Fig. 1a), our bottom-up estimate would be around 11.8 Gg yr⁻¹ in 2014. Based on these considerations, we suggest that the growth in top-down emissions between 2012 and 2014 can be explained by new emissions from newly installed, at least partly unabated, HCFC-22 production capacity, combined with the “switching off” of some, but not all, abatement that was installed during the CDM period (consistent with observations from one previous study in India [14]).”

How do we know that the representative and transport errors of a discretely sampled gas are not simply smoothing the inferred emission changes in the inverse model?

See Figure R1, where we show a forward model run based on emissions from the “Developing Country With Abatement” global estimate. Since the AGAGE 12-box model has some estimate of the large-scale mixing timescales in the atmosphere, we expect the predicted mole fractions to broadly reflect any smoothing of the emissions signal that would be seen by the atmospheric data. The forward model run shows a decline in growth rate, and simultaneous collapse of the inter-hemispheric difference after 2015. Neither of these features are reflected in the data. After 2015, the predicted mole fractions differ from the data by many times the estimated model-data mismatch uncertainty (notwithstanding any small, but unaccounted-for, errors associated with the use of inter-annually repeating transport, as discussed below).

This simulation shows that the atmospheric data would be strongly influenced by such a large change in emissions. In order for the inversion to match the data within uncertainties would require it to also show a substantial decline in emissions, which it does not.

We have included this figure in the Supplement (Fig. S1), and now refer to it in the main text in the “Evaluating global emissions using atmospheric observations” subsection on lines 86 – 90:

“The AGAGE data show a renewed increase in the HFC-23 growth rate from 2016, which reached 1.1 ± 0.05 pmol mol⁻¹ yr⁻¹ in 2018, when global annual mean mole fractions were 31.1 parts per trillion (pmol mol⁻¹; Fig. 2). In contrast, a forward model run using our “Developing Country With Abatement” estimate suggests that the global mean mole fraction growth rate should have declined to less than zero after around 2016 (Supplement Fig. S1).”

Figure R1: AGAGE 12-box model mole fractions predicted from the “Developing Country With Abatement” global bottom-up emissions estimate (solid lines). Data are shown as dots (archive) or points with error bars (in situ, see Figure 2 in main text). An offset was applied to the model (primarily accounting for the influence of pre-1990 emissions) so that the 2014 annual mean emissions agree with the data.

[1] Montzka, S. A. et al. An unexpected and persistent increase in global emissions of ozone-depleting CFC-11. *Nature*. 557, 413–417 (2018).

[2] Prinn, R. G. et al. History of chemically and radiatively important atmospheric gases from the Advanced Global Atmospheric Gases Experiment (AGAGE). *Earth Syst. Sci. Data* 10, 985 (2018).

[3] Rigby, M. et al. Increase in CFC-11 emissions from eastern China based on atmospheric observations. *Nature*. 569, 546–550 (2019).

Why did the top-down and bottom-up methods decouple in 2012, before the self-reported large reductions in China and India begin?

We have addressed this in the first response to reviewer #3 in adding extra information into the paragraph that discusses top-down and bottom up between 2012 and 2015 on lines 99 to 108 in the revised manuscript, which now reads:

“After 2012, our top-down estimates grow more slowly than would be expected from HCFC-22 production and the decline in reported CDM abatement (Fig. 1a); CDM abatement declines to zero by the end of 2014, resulting in our “abatement” and “no-abatement” bottom-up estimates nearly converging at 20.8 Gg yr⁻¹, compared to the top-down estimate of 14.5 Gg yr⁻¹. This corresponds to a maximum discrepancy between top-down and bottom-up with “abatement” of 6.3 Gg yr⁻¹ in 2014. If the full abatement capacity installed during the CDM had continued, our

bottom-up estimate would be substantially lower in 2014; if we assume that the maximum amount abated during the CDM (9.0 Gg yr^{-1} in 2011) had continued (red dotted line in Fig. 1a), our bottom-up estimate would be around 11.8 Gg yr^{-1} in 2014. Based on these considerations, we suggest that the growth in top-down emissions between 2012 and 2014 can be explained by new emissions from newly installed, at least partly unabated, HCFC-22 production capacity, combined with the “switching off” of some, but not all, abatement that was installed during the CDM period (consistent with observations from one previous study in India [14]).”

How is the 1-sigma bars exactly determined and what does it encapsulate? In particular, if the model reuses the same meteorology each year — as per the Methods section — how does real-world interannual variability in transport (such as might be expected with the transition from a La Niña state to strong El Niño state during this period) influence the results?

As stated in the methods (lines 224 – 226 of the original manuscript; lines 164 – 167 of the revised manuscript), the 1-sigma error bars “incorporate uncertainties due to the prior constraints, measurements (repeatability and calibration scale propagation), model representations of the data (assumed to be equal to the monthly baseline variability) and the lifetime of HFC-23”. To add clarification, these extra lines have been included in the caption to Figure 1a (page 4 of the revised manuscript).

The reviewer is correct that the error due to the use of interannually varying meteorology has not been included, as it is difficult to assess without 3D model runs. Previous studies that have compared 3D and box model estimates of global emissions for other species have shown differences of only a few percent (e.g. Saikawa, et al.[1] for HCFC-22). Therefore, we expect this uncertainty to be much smaller than the magnitude of the signals we are investigating in this paper. We have added the following lines (169 – 172 in the revised manuscript to the methods section:

“The uncertainty in the derived global, annual emissions introduced by using inter-annually repeating meteorology could not be assessed. However, previous comparisons of box model and three-dimensional top-down emissions estimates for other species suggests that this source of uncertainty will be substantially smaller than the magnitude of the emissions changes that are the subject of this paper[28]”

[1] Saikawa, E. et al. Global and regional emission estimates for HCFC-22. *Atmos. chem. Phys.* 12, 10033–10050 (2012)

As a lesser point, I find it unusual for a top-down evaluation study that there are no observations shown in the main text. I would move the present Figure 3 into the supplement (the five locations are already described in the text with words), and a version of Figure S1 into the main text.

As the manuscript was focused more on HFC-23 emissions, the authors thought that observations were best presented in the Supplement. However, we realize that it is unusual and therefore have included Figure S1 in the manuscript, which has now become Fig. 2 (page 5 of revised manuscript). Entries in the text that reference the observation plot have been adjusted accordingly.

Figure 3 from the original manuscript (plot of site locations) has been moved into the Supplement and is now Figure S2. Entries in the text that reference the location plot have been adjusted accordingly.

Minor Comments

L19 and L226 - Why does the manuscript use two different HFC-23 lifetimes?

The lifetime is calculated within the 12-box model, based on the Burkholder et al.[1] OH rate constant, SPARC stratospheric lifetime and the box model atmospheric transport. Therefore, the box model lifetime differs very slightly from the “accepted” lifetime. The differences are extremely small and will have only negligible impact on the outcome.

[1] Burkholder, J. et al. Chemical Kinetics and Photochemical Data for Use in Atmospheric Studies—Evaluation Number 18. NASA panel for data evaluation technical report (2015).

Fig. 1 caption - “pot-2015” should be “post-2015”

Many thanks for pointing out this type, it has been corrected in the revised manuscript (page 4).

I find Figure 1 overly confusing for the information it is trying to convey (although, this is personal preference). Fig. 1a includes the red-dotted line that is never discussed in the text anywhere aside from the caption, and its purpose for being shown is not made clear; I think it could be cut. I understand the purpose of the hatching but personally think it would be better conveyed using simple vertical lines and maybe rectangular background shading to delineate the temporal periods of interest (pre-CDM abatement vs. CDM period vs. post-2015 reductions). The thing that matters most here is the comparison of the top-down emissions (blue line) vs. the reported emissions (green line). To that end, the dots showing the developed vs. developing nations non-abatement inventories seem superfluous and might be best relegated to the Supplement. Fig. 1b would convey the same information more simply as a stacked area chart.

We disagree with the comment about the dotted red line, as it is discussed on lines 91-92 in original manuscript (page 5, lines 102 – 105 in the revised manuscript). With regards to the developed vs. developing countries HFC-23 bottom-up emission estimates (shown as dotted lines in Fig.1a), we feel that it is important to show this data in conjunction with the totals as it demonstrates the changing trends between the dominance of emissions by developed vs. developing countries over time, especially towards the end of the timeseries when developing countries emit the most HFC-23.

As the reviewer has stated that this comment is a personal preference, and that the other two reviewers have not caused issue with the figure, we would like to keep the plot as it is. However, if the editor would like us to incorporate any of these recommended changes, we will do so willingly.

Fig. S1 caption – dashed lines presumably show latitude band growth rates, not hemispheric growth rates. The dashed lines in Figure S1b (now Figure 2b, page 5, in the revised manuscript) represent the semi-hemispheric growth rates for HFC-23. The figure caption has been altered to show this and now reads:

“Bottom panel: shows model-derived annual HFC-23 growth rates (global – blue solid line with 1-sigma uncertainty indicated by shading; dashed lines show semi-hemispheric growth rates) in $\text{pmol mol}^{-1} \text{ yr}^{-1}$.”

REVIEWERS' COMMENTS:

Reviewer #1 (Remarks to the Author):

The authors have reasonably incorporated my comments in the revised version of the article.

Reviewer #2 (Remarks to the Author):

I'm satisfied that my reviewer comments have been dealt with carefully and thoroughly, and am happy to recommend publication. This is important work and a very well put together manuscript.

Reviewer #3 (Remarks to the Author):

In general, the authors have adequately addressed my concerns. I have two remaining comments for the editor's discretion. First, in the interest of the scientific method, every article should be able to be reproduced based on its description alone. I do not believe the description of the error assumptions reported for the inverse model would be adequate for me to reproduce the simulations without contacting the authors. Second, I would never personally submit a figure as complicated as Fig. 1. As a close reviewer, it took me several reads to try to figure out what important information was being conveyed in the context of the results. As I originally stated, this is a matter of personal preference, but I would encourage the authors to consider the "less is more" mantra for future publications, and relegate secondary information to the supplemental materials.

We thank the reviewers for their thoughtful comments on our revised manuscript. Below, we respond to each of the comments in turn, with our responses are in red.

Reviewer #1 (Remarks to the Author):

The authors have reasonably incorporated my comments in the revised version of the article.
Thank you.

Reviewer #2 (Remarks to the Author):

I'm satisfied that my reviewer comments have been dealt with carefully and thoroughly, and am happy to recommend publication. This is important work and a very well put together manuscript.
We thank reviewer #2 for their kind comments.

Reviewer #3 (Remarks to the Author):

In general, the authors have adequately addressed my concerns. I have two remaining comments for the editor's discretion. First, in the interest of the scientific method, every article should be able to be reproduced based on its description alone. I do not believe the description of the error assumptions reported for the inverse model would be adequate for me to reproduce the simulations without contacting the authors.

We have tried to clarify the description of our error assumptions reported for the inverse model used in this manuscript to enable reproducibility of simulations without having to contact any of the authors. We have changed the methods section between lines 162 to 174 to read as follows:

"The lifetime of HFC-23 was approximately 240 years in the model, based on the HFC-23 - hydroxyl reaction rate estimated by Burkholder et al.[30] and stratospheric lifetime from Ko et al.[4]. The model uses annually repeating meteorology and a hydroxyl radical concentration climatology[31], tuned to match the growth rate of methyl chloroform[28]. Emissions were inferred using a Bayesian method in which the emissions growth rate was weakly constrained a priori[27, 32]; a prior emissions growth rate of zero plus or minus 20% of the maximum bottom-up emissions from Miller et al.[8] was assumed (i.e. a priori emissions growth of $0 \pm 2.7 \text{ Gg yr}^{-1}$). In the inversion, uncertainties on the measurements were assumed to be equal to the monthly baseline variability for the high-frequency in situ data (thereby incorporating measurement repeatability and a model representation error term related to the sub-monthly timescales not resolved by the model). Uncertainties on the archived data points were assumed equal to the quadratic sum of the repeatability and the mean variability during the in situ data period, scaled by the mole fraction difference between the archive and in situ data (also as an approximation of the model representation error for those archive samples). Following the Bayesian method in Rigby et al.[32], the posterior uncertainties are dependent on these assumed prior and measurement uncertainties, and, following Rigby, et al.[27], are augmented with a term related to systematic uncertainties in the calibration scale (approximately 3%) and loss frequency (approximately 21% [4])."

Second, I would never personally submit a figure as complicated as Fig. 1. As a close reviewer, it took me several reads to try to figure out what important information was being conveyed in the context of the results. As I originally stated, this is a matter of personal preference, but I would encourage the authors to consider the "less is more" mantra for future publications, and relegate secondary information to the supplemental materials.

We thank the reviewer for their comment and will try to use the "less is more" mantra for future publications.